

# Different photosynthetic adaptation of *Zoysia* spp. under shading: shade avoidance and shade tolerance response

Xiao Xu, Hongli Wang, Guangyang Wang, Xiaoning Li, Xiaoyan Liu and Jinmin Fu

Coastal Salinity Tolerant Grass Engineering and Technology Research Center, Ludong University, Yantai, China

## ABSTRACT

Reduction of ambient solar radiation is an important external challenge for plants, which affects photosynthesis and morphogenesis in agroforestry or gardening. As bottomed sessile organisms, turfgrasses have a set of sophisticated photosynthetic strategies to survive and deal with this abiotic stress. Zoysiagrass (*Zoysia* spp. Willd.) is an important warm-season, perennial turfgrass that tolerates adversity, wear, trampling and extensive management. However, whole photosynthetic characteristics reaction of the zoysiagrass to shade stress have not been described because our knowledge in this area is very limited. In this study, 85% shade treatment was applied to nineteen zoysiagrass genotypes, and morphological observations and extensive determinations on plant heights, photosynthetic pigments, fluorescence dynamic curves among other parameters were made. The results showed that vegetal and photosynthetic responses of zoysiagrass were affected by shade treatment to varying degrees. Further analysis based on the principal component, subordinate function analysis and clustering methodology revealed that different shading response strategies were adopted by zoysia under shade surroundings. They were divided into four categories. The strongest shade-avoidant response strategy was adopted by 'ZG48' and 'WZG59', which had the largest comprehensive evaluation (D) values, and the stabilized shade-tolerant response was taken on by 'ZG-3' and 'ZG64', which had the lowest D values. Other varieties applied a medium strategy but with a certain tendency. These findings provide new insights into different shading response tactics of turfgrass: shade avoidance and shade tolerance response, which could be selected for further elucidation of the molecular mechanism of plant adaptation to shade environments.

Corresponding author
Jinmin Fu, turfcn@qq.com, jfu@ldu.edu.cn

## INTRODUCTION

Due to the diurnal variation of sunlight, the movement of clouds, crown canopies and crowded buildings and such the daily photon flux available for plants is frequently reduced (*Ruban, 2009*). Although many studies of shade tolerance and avoidance variation are mainly on the model plant, *Arabidopsis*, its disadvantage is that it is not a naturally existing understory and cannot be authentically considered a shade-tolerant plant (*Gommers et*

al., 2013). Natural shade-tolerant plant communities need further research (*Warnasooriya & Brutnell, 2014*; *Roig-Villanova & Martinez-Garcia, 2016*). Lawns are highly approbatory and largely prefabricated landscape design elements across the globe. As crucial ecological barriers, they can beautify the environment, protect ecology, provide recreation places, improve the regional microclimate, and alleviate the heat island effect (*Ignatieva et al., 2020*). Zoysiagrass (*Zoysia* spp. Willd.) is a warm-season perennial turfgrass with strong resistance to heat, drought and abrasion caused by traffic and soil compaction and relative resistance to shade and salinity compared to other turfgrasses (*Harivandi et al., 1984*), which are extensively used in lawn construction, such as ornamental, recreational, playground and courtyard applications throughout the southern humid region to the northern transition zone. Additionally, zoysiagrass is rich in germplasm resources, including 11 species (varieties), each with large genetic variations for shade tolerance traits among them (*Sladek, Henry & Auld, 2009*; *Liu et al., 2019a*; *Liu et al., 2019b*).

It is estimated that 20% to 25% of grass in the U.S. and 50% of turf in China are growing in shaded areas (*Jiang, Duncan & Carrow, 2004*; *Xu et al., 2013*). Light is necessary for plant photosynthesis and morphogenesis, but the solar radiation received by turfgrass under shaded environments is limited. The photosynthetic photon flux (PPF) measurements of turfgrass growing under deciduous shade, coniferous shade and building shade compared with full sun in nature were reduced by 50.8%, 82.3% and 72%, respectively (*Bell, Danneberger & McMahon, 2000*). In response, there are varying degrees of changes in the morphology and physiology of turfgrass species because of their intraspecific and interspecific variations occurring in different ecological habitats (*Gardner & Goss, 2013*). To survive under shading, most turfgrass species exhibit stem and leaf elongation, narrower leaf blades, a reduction in leaf area, longer internodes, and depressed photosynthetic efficiency, but this can result in gradually reduced turf quality and coverage, along with vulnerability to diseases and pests and recession (*Sladek, Henry & Auld, 2009*; *Malik et al., 2014*). Therefore, many shade-tolerant turfgrasses are selected with weak elongation response, small increases in leaf area and negligible etiolation, but with some physiological characteristics changed (*Chhetri, 2017*; *Taylor, 2019*; *Petrella & Watkins, 2020*). In brief, these responses to shade can be used as parameters to constantly select shade-tolerant turfgrass for modern landscape design. For example, *Patton (2010)* estimated the turf performance index (TPI) values of 38 zoysia resources under 90% shade, which were generated for ranking the cultivars that represented the number of times that germoplasm occurred in the top statistical group across coverage, quality, color, and density. The results showed that 'Diamond' and 'Zorro', belonging to *Zoysia matrella* cultivars, had the highest rating, and 'Belair' and 'Meyer', belonging to *Zoysia japonica* cultivars, had the poorest shade tolerance. *Wherley et al. (2011)* evaluated the performance of ten zoysia cultivars under 89% shaded environments for quality, density, color, vertical canopy height, and extent of lateral spread, and the results suggested that three *Z. matrella* cultivars 'Royal', 'Zorro' and 'Shadow Turf' were ranked in the top statistical grouping under heavily shaded environments. However, there are few studies on zoysia photosynthetic characteristics under shading stress.

Light is a form of energy that drives numerous life processes in plants due to its transformation of energy from chemical bonds. Light energy absorbed by chlorophylls associated with photosystem II (PSII) could be used to drive the primary photochemical reaction, in which an electron ($e^-$) is transferred from the reaction center chlorophyll, P680, to the primary quinone acceptor of PSII, QA. Simultaneously, absorbed light energy is lost from PSII as chlorophyll fluorescence or heat energy (*Baker, 2008*). Shade can affect the primary reactions of photosynthesis, electron transfer, photosynthetic phosphorylation and carbon assimilation and eventually lead to a reduction in photosynthetic efficiency (*Taiz & Zeiger, 2010*; *Gommers et al., 2013*; *Mathur, Jain & Jajoo, 2018*). The measurement of chlorophyll fluorescence is now utilized widely to evaluate changes in the photochemistry of PSII, photosynthetic efficiency, reaction center, electron transfer from PSII to the acceptor side of PSI in the intersystem chain, and so on by analyzing the output fluorescence parameters (*Kalaji et al., 2016*). It has been verified to be a valuable technique to screen and estimate resistance response levels for rapid perturbations in photosynthesis.

To date, several studies on the photosynthetic response mechanism of turfgrass to shade stress have been rarely reported, so they have been conducted with zoysiagrass. Whether photosynthetic indices of zoysiagrass under shade conditions could be applied in quickly analyzing shade response is worth exploring. In this research, we made morphological observations, accompanied by extensive measurements of photosynthetic pigments and chlorophyll fluorescence under sunlight and artificial shade conditions to obtain valuable information about the primary photosystem stoichiometry of nineteen zoysiagrass. Comprehensive evaluations identified four categories of zoysia germplasms, the strongest shade-avoidance response, the moderate shade-avoidance response, the mid-degree shade-tolerant response, and the most powerful shade-tolerant response strategy, by principal component analysis (PCA), subordinate function analysis and clustering methodology. These analyses subsequently laid a foundation for the molecular response mechanisms.

## MATERIAL AND METHODS

### Plant material and shade treatment

The study was carried out on nineteen zoysiagrass accessions including 'ZG-3', 'Wuhao-1', 'WZG99', 'ZG63', 'Manila', 'ZG31', 'Nanling', 'ZG45', 'WZG55', 'WZG59', 'ZG66', 'ZG65', 'ZG67', 'WZGF8', 'WZG91', 'WZG97', 'ZG64', 'WZG85' and 'ZG48' (origin regions are shown in Table S1). Zoysiagrasses with similar areas and weights were dug out with a ring knife from the germplasm resource garden in the Coastal Salinity Tolerant Grass Engineering and Technology Research Center, Ludong University. Then they were planted into 6.6-cm-diameter and 26-cm-deep plastic pipes filled with cultivated organic soil in a greenhouse, with a day/night temperature of 28/20 $\pm$ 2 °C (day/night), 60% relative humidity and 738 $\mu$mol m$^{-2}$ s$^{-1}$ of photosynthetically active radiation on average. The temperature and humidity were obtained with a digital hygrometer thermometer. Photosynthetically active radiation values were measured by in-situ measurement approach using a hand-held illumination photometer at 11 a.m. on a clear day for three biological repetitions. The grasses were irrigated with 1/2-strength Hoagland's solution and clipped once a week until growth was consistent after 30 days.

Then, shade treatments (other than natural sunlight) were applied using a polyethylene black shade cloth by constructing a shade structure surrounding the established plots in the greenhouse. Each pipe represents one replication, *i.e.*, there were three replications under shade and three repeats under full sunlight per accession. Photosynthetically active radiation measurements on shaded lawns averaged 110 $\mu$mol m$^{-2}$ s$^{-1}$ at 11 a.m., and the shade degree was 85% of full sunlight in the greenhouse. During this period, the grasses were irrigated with nutrient solution weekly and watered twice a week continuously. Shade cloth was removed after 50 days, when there were classic shade-avoidance syndrome differences in many accessions (*Gommers et al., 2013*). All measurements were sampled or measured within 2 days under the controlled conditions above.

## Measurement of morphological parameters

Growths of zoysiagrass under shade treatment and natural light were observed and compared. Plant height was measured from the substrate surface of each tube to the top in four directions using a ruler. The average value of the three tubes was used as the plant height of the zoysiagrass.

## Determination of photosynthetic pigments

Photosynthetic pigment contents were calculated according to the methods described by *Gratani (1992)* with some changes. In brief, the photosynthetic pigments were extracted from 0.1 g leaf samples, taken from the top 2nd–3rd leaves of zoysiagrass under shaded and sunlight conditions. Then, 10 ml dimethyl sulfoxide was added and incubated in the dark for 72 h. Then, the absorbance of the extracting solution was measured at 663, 645 and 440 nm using a spectrophotometer. The photosynthetic pigments concentrations, including chlorophyll *a* (Chl *a*), chlorophyll *b* (Chl *b*), total chlorophyll (Chls) and carotenoid (Caro), were calculated by following formulas with three biological repetitions.

$$\text{Chl}a \ (\text{mg g}^{-1}) = [12.72 \times (\text{OD}663) - 2.59 \times (\text{OD}645)] \times \text{V/W}$$
$$\text{Chl}b \ (\text{mg g}-1) = [22.88 \times (\text{OD}645) - 4.67 \times (\text{OD}663)] \times \text{V/W}$$
$$\text{Chls} \ (\text{mg g}^{-1}) = \text{Chl}a + \text{Chl}b$$
$$\text{Caro} \ (\text{mg g}^{-1}) = [4.7 \times (\text{OD } 440) - 0.27 \times (\text{Chl}a + \text{Chl}b)] \times \text{V/W}$$

where, V = Volume of Extract (L), W = weights of Fresh leaves (g), OD = optimal density.

## Chlorophyll fluorescence measurements

A pulse-amplitude modulation (PAM) fluorometer (PAM 2500; Heinz Walz GmbH, Germany) was used to measure changes in chlorophyll fluorescence. First, the top 2nd–3rd leaves of shaded and unshaded zoysiagrass were dark-adapted for 20 min to close all the PSII reaction centers and acquire the maximal fluorescence intensity. Then, they were exposed to 3,000 mmol photons m$^{-2}$ s$^{-1}$ red light conditions to generate the OJIP transients and chlorophyll fluorescence kinetics. The lowest fluorescence when exposed to light was defined as the O point, while the highest fluorescence was defined as the P point. The rapid chlorophyll fluorescence induction kinetic curve referred to the fluorescence changes from the O point to the P point, which was mainly related to the initial photochemical

reaction of PSII. Based on the theory of energy fluxes in biofilms, the OJIP tests can further translate the primary data into other biophysical parameters and be used to calculate a series of indices according to *Baker (2008)* and *Zivcak et al. (2014)*. At least three replicates per treatment were randomly selected for each zoysia. The chlorophyll fluorescence kinetic curve was analyzed and plotted with Origin 9.0 software.

## Statistics and analysis

The shade response coefficients were calculated using each 85% shade index divided by the indicator in light. Correlation among the eighteen screening indices was determined using Spearman analysis in IBM SPSS version 19.0 software. Additionally, to investigate the response patterns of photosynthetic physiological variations at the 85% shade level, PCA was performed, which reduced the mass of multidimensional datasets to a few informative groups to identify the key indicators. The index weight of the principal components was calculated as $W_j = E_j / \Sigma E_j$ ($j = 1, 2,\ldots\ldots, n$), where $E_j$ indicates the jth eigenvalues. Subordinate function values were reckoned with $U_j = (Y_j - Y_{min})/(Y_{max} - Y_{min})$ ($j = 1, 2,\ldots\ldots, n$), where $Y_j$ indicates the score of the jth principal component index, $Y_{min}$ indicates the minimum score of the corresponding principal component index, and $Y_{max}$ indicates the maximum score of the corresponding principal component index. Comprehensive evaluation values (D) of different zoysiagrass accessions were computed as $D = \Sigma(U_j \times W_j)$ ($j = 1, 2,\ldots\ldots, n$), which indicates the shade response degree exposed to 85% shading stress (*Jia et al., 2020*; *Liu et al., 2019a*; *Liu et al., 2019b*). Clustering analysis of these nineteen different zoysiagrass accessions based on D values. The distance between D values was measured by the nearest neighbor element analysis model for IBM SPSS Statistics, according to the squared Euclidean distance.

The results are presented as the mean values of at least three independent biological replicates. The calculated averages were used to create bar charts with Origin Pro 9. Error bars in figures represent standard deviations calculated from triplicates. The asterisks, * ($P < 0.05$) and ** ($P < 0.01$), indicate significant differences obtained through IBM SPSS software by the independent samples *T*-test, when indicators in shade were compared with those in light, respectively.

## RESULTS

### Effects of shade treatment on phenotypes of zoysiagrass

After shade treatment (50 days), compared to their control, morphological differences were observed among nineteen zoysiagrass (Fig. 1). Some zoysiagrass did not show visible changes in turf quality, exerting a shade-tolerant response, *e.g.*, 'ZG-3', 'Wuhao-1', 'WZG99', 'ZG63', 'Manila', and 'ZG31'. Some others, such as 'ZG48', 'WZG85', 'WZG97', 'WZG91', and 'WZGF8' were obviously changed, including plant height increase, stem elongation and thinning, as well as easier lodging, which reflected a shade-avoidance response. Still others had moderate variation, for example, 'Nanling', 'ZG45', 'WZG55', 'WZG59', 'ZG66', 'ZG65', and 'ZG67'. Analogously, surveys of plant height showed that 'ZG48', 'WZG85', 'WZG97', 'WZG91', 'WZGF8', and 'ZG65' increased significantly, 'ZG64' decreased markedly, while other varieties had no noteworthy differences (Fig. 2).

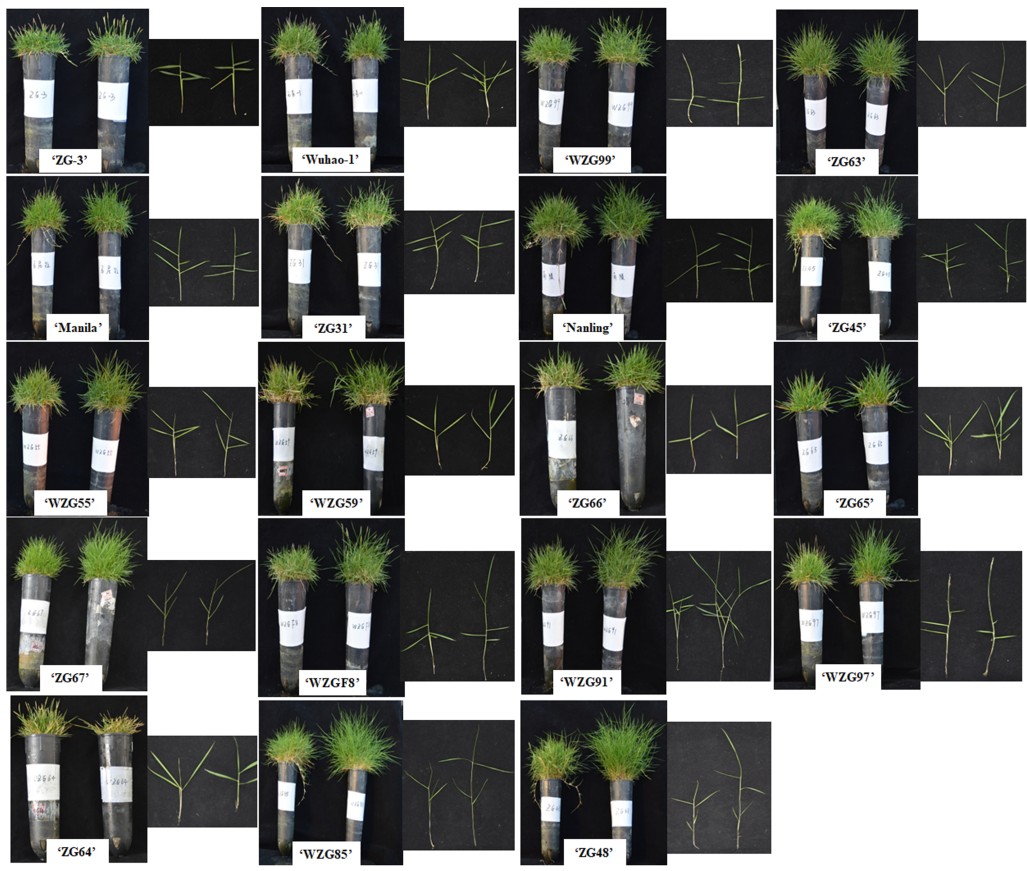

**Figure 1** **Morphological phenotypes of zoysiagrass after 85% shade treatment.** Accession that grew in light was in the left side of each picture and the shading one was in the right side.

## Effects of shade treatment on photosynthetic pigment contents and compositions in leaves of zoysiagrass

All zoysiagrass species contained Chl *a*, Chl *b*, and Caro in leaves, and the contents of individual photosynthetic pigments differed. Shade significantly augmented pigment contents to capture more light energy in 'Wuhao-1', 'WZG99', 'Manila', 'ZG31', 'Nanling', 'ZG45', 'WZG55', 'WZG59', 'ZG66', 'ZG65', 'ZG67', 'WZGF8', 'WZG91', 'WZG97', 'WZG85', and 'ZG48' while there were nearly no changes in 'ZG-3' and 'ZG64'. In addition, the pigment contents in 'ZG63' decreased markedly (Figs. 3A–3D). Among them, the Chl *a* contents of 'WZG55', 'WZG59', 'ZG66', 'ZG67', 'WZG97' and 'WZG85' increased by more than 50%, the Chl *b* contents of 'WZG99', 'Nanling', 'WZG55', 'WZG59', 'ZG67', and 'WZG97' increased by greater than 50%, and the Caro contents of 'WZG59', 'ZG67', and 'WZG97' increased by more than 50%. Shade also changed the compositions of photosynthetic pigments, which reflected the ratios of Chl *a*/Chl *b* and Chls/Caro (Figs. 3E–3F). The variational degrees of Chl *a* and Chl *b* in 'ZG-3', 'WZG99', 'Manila', 'ZG31', 'ZG45', 'WZG55', 'WZG59', 'ZG67', 'WZGF8', 'WZG91' and 'ZG64' were consistent, so the Chl *a*/Chl *b* ratio was not significant in response to shade treatment. However,

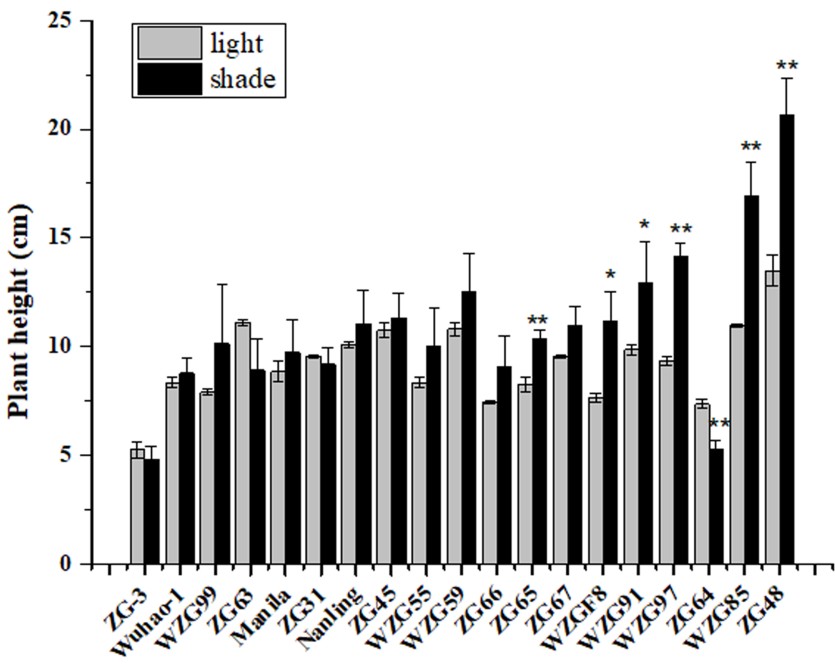

**Figure 2** **Plant height of zoysiagrass after shade treatment.** Asterisks (* and **) indicate statistically significant differences at $P < 0.05$ and $P < 0.01$ respectively, when indicators in shade were compared with those in light, by SPSS version 19.0 software at the independent samples $T$-test.

there were some inconsonant Chl *a*/Chl *b* ratios; for example, 'Wuhao-1' and 'Nanling' decreased notably, whereas 'ZG66', 'ZG65', 'WZG97', 'ZG48', and 'WZG85' increased. Meanwhile, Caro became less abundant relative to Chls under shade in most zoysiagrass, such as 'Wuhao-1', 'Manila', 'ZG31', 'Nanling', 'ZG45', 'WZG55', 'WZG59', 'ZG66', 'ZG65', 'ZG67', 'WZGF8', 'WZG91', 'WZG97', 'ZG64', 'WZG85', and 'ZG48. Only 'ZG-3', 'WZG99', and 'ZG63 persisted in no significant changes.

## Effects of shade treatment on chlorophyll fluorescence of zoysiagrass

The impacts of low light treatment on the original photochemical activity of PSII were determined through the chlorophyll fluorescence transient-JIP test, which is a powerful tool to reflect the state of the photosynthetic apparatus. Chlorophyll fluorescence kinetic curves of nineteen zoysia grasses were elevated to different degrees induced by shade and thus difficult to sort directly (Fig. 4). The increase occurred significantly in steps J to P, which involved multiple QA$^-$ flows and promoted the reduction process of electron transfer to the acceptor side of PSI, especially in 'ZG48'. However, there might be the fewest changes in 'ZG-3' suggesting the tolerance of the photosystem to the shading environment.

Chlorophyll fluorescence kinetic parameters could reflect plentiful photosynthesis information, such as absorption, capture, transfer and distribution of light energy, activity of the reaction center, photosynthetic efficiency and performance. We obtained a large number of PSII photosynthetic parameters under shaded and sunlight conditions, including fluorescence parameters derived from the extracted data (Fo, Fm), yield or flux

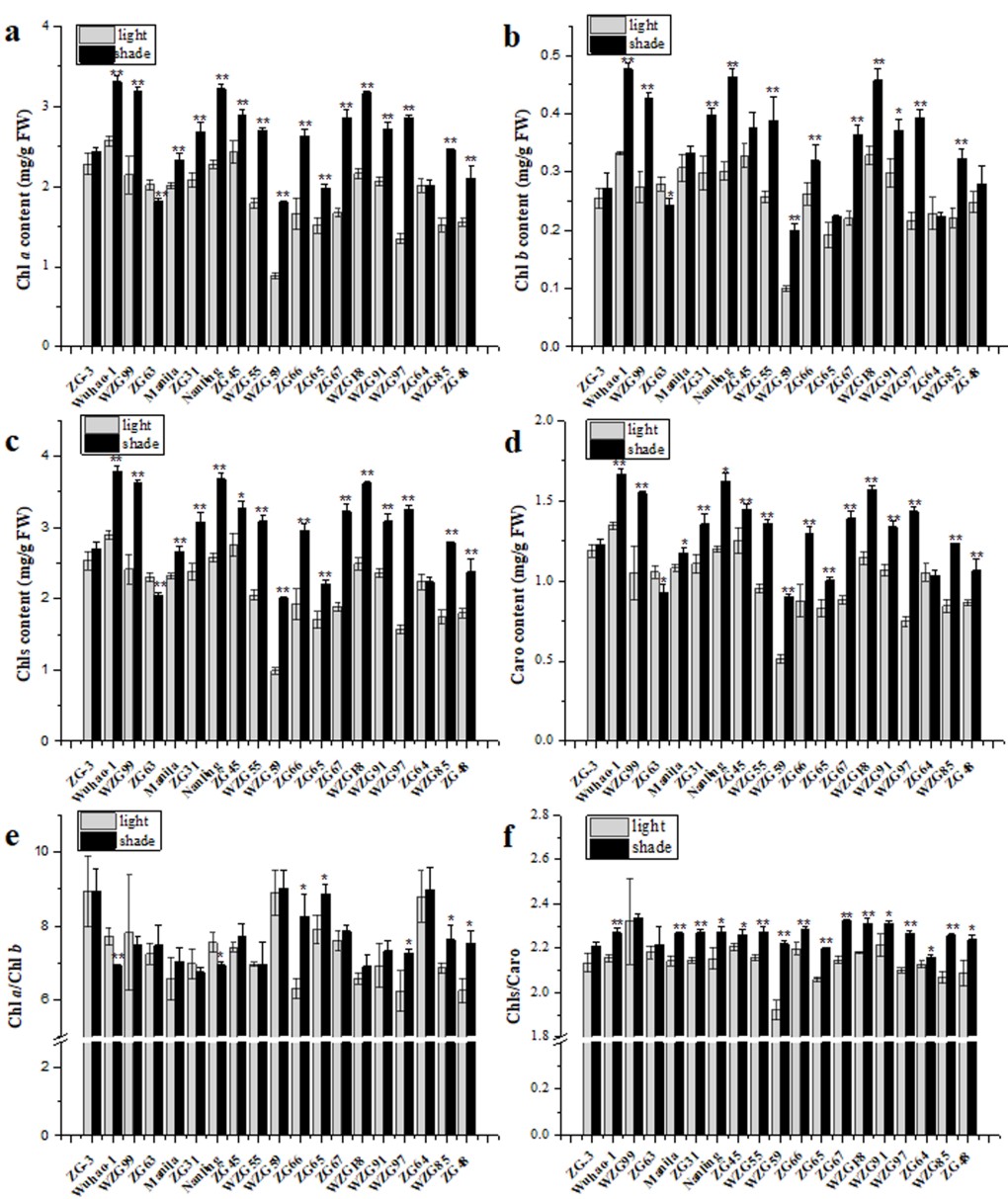

**Figure 3  Changes of photosynthetic pigments in zoysiagrass including the contents of Chl *a* (A), Chl *b* (B), total Chls (C), Caro (D) and the ratios of Chl *a*/Chl *b* (E), Chls/Caro (F).** Asterisks (* and **) indicate statistically significant differences at $P < 0.05$ and $P < 0.01$ respectively, when indicators in shade were compared with those in light, by SPSS version 19.0 software at the independent samples $T$-test.

ratio parameters ($\varphi$Po, $\psi$Eo, $\varphi$Eo), specific energy fluxes per reaction center (ABS/RC, TRo/RC, ETo/RC, DIo/RC), and performance indices (PI$_{ABS}$, PI$_{CS}$, PI$_{total}$) (Table S2). After shade treatment, photosynthetic parameter levels changed differently in diverse zoysiagrass species. Of them, maximal fluorescence yields (Fm) were essentially enhanced in most germplasms, except for 'ZG-3', 'WZG99', 'ZG63', and 'ZG31' (Fig. 5A). The ratio of $\varphi$Po could be used to estimate the maximum quantum efficiencies of PSII. The

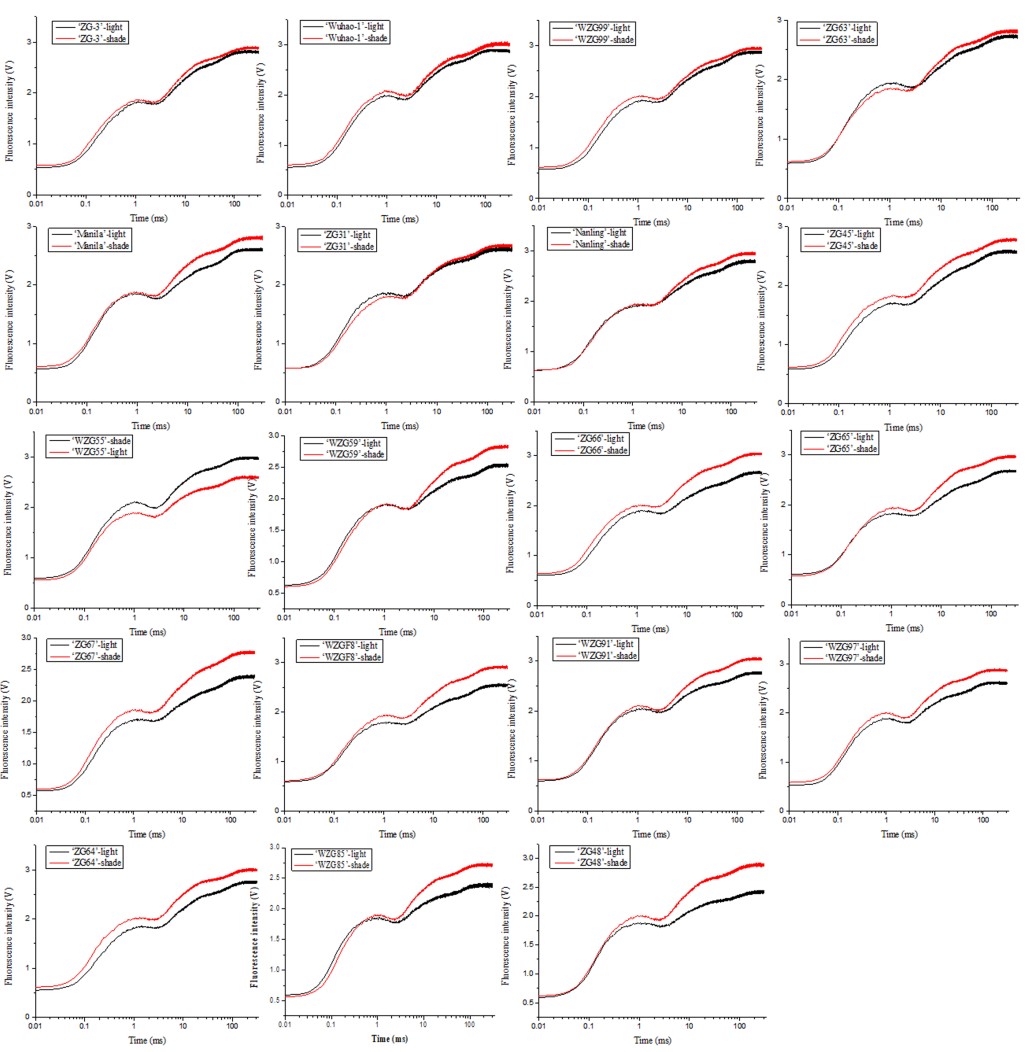

**Figure 4 Changes of chlorophyll fluorescence kinetic curves of zoysiagrass after shading.** The curves were plotted using OriginPro 9.0 software.

results showed that 'WZG55', 'WZG59', 'ZG65', 'ZG67', 'WZG91', and 'ZG48' were markedly elevated, while other accessions, 'ZG-3', 'Wuhao-1', 'WZG99', 'ZG63' etc., had no significant differences (Fig. 5B). Additionally, the light energy absorbed by the PSII unit reaction center (ABS/RC) and the captured light energy used for reduction $QA^-$ (TRo/RC) were dramatically increased in 'ZG-3', 'WZG99', 'ZG31', 'WZG55', 'WZG59', 'ZG64', and 'ZG48' the electron transport (ETo/RC) was notably boosted in 'Manila', 'WZG55', 'ZG66', 'ZG67', 'WZG91', 'WZG97' and 'ZG64' and the energy dissipated (DIo/RC) was conspicuous in 'ZG-3', 'ZG31', 'WZG55', 'WZG59', 'ZG65', 'ZG67', and 'ZG48' (Figs. 5C–5F). In addition, performance indices based on absorbed light energy (PI$_{ABS}$) and unit cross-sectional area (PI$_{CS}$), which are more sensitive to changes in the photosynthetic apparatus, were enhanced in 'Manila', 'ZG31', 'Nanling', 'WZG55', 'WZG59', 'ZG66', 'ZG65', 'ZG67', 'WZGF8', 'WZG91', 'WZG97', and 'ZG48' (Figs. 5G–5H). We observed

that some cultivars ('ZG-3', 'Wuhao-1', 'WZG99') had relatively high-performance values in Fm, $\varphi$Po, $\psi$Eo, $\varphi$Eo, ETo/RC, $PI_{ABS}$, $PI_{CS}$ and $PI_{total}$ under light, whose variation ranges were not large after shading, while others ('ZG48', 'WZG97', 'WZG59') increased significantly to elevate photosynthetic efficiency and capacity whose indicators in light were correspondingly low, but most of them did not reach the levels above. In conclusion, a large number of indicators were obtained, but their photosynthetic strategies could not be well determined.

## Correlation and principal component analysis among the characteristics of zoysiagrass

The units and ranges of different indices were disparate, causing confusion in the comparative analysis. To avoid the influence of different dimensions, the shading response coefficients of each test index were calculated (Table S3). From these, we found that the changes varied for each single coefficient in nineteen zoysia accessions; some indices were greater than 1, and others were less than 1. The correlation analysis of shade response coefficients showed that each test index made a different contribution to the shading condition. There were significant positive or negative correlations among these indices, indicating that the shade response information was overlapped (Table 1). Therefore, other statistical methods should be applied to comprehensive assessment.

To reduce the overlap and make up for the deficiency of index evaluation, a PCA was implemented to integrate these eighteen shading response coefficients (Table 2). The Kaiser–Meyer–Olkin (KMO) value was 0.66, and the significant chi-square test was 0.00, indicating a correspondingly high pertinence in each indicator and signifying that factor analysis was appropriate for use in this paper. We obtained five new principal components with eigenvalues greater than 1.0, whose variance contribution rates were 44.88%, 16.66%, 13.40%, 7.86% and 6.35%, and their cumulative contribution rate was 89.16%. Thus, eighteen independent photosynthetic variables were transformed into five principal factors, which could be used to measure zoysia shading responses.

Furthermore, the eigenvector coefficients of the five principal components were obtained by the original component matrix divided by SQRT (E), respectively. Expressions for Y1 ~Y5 (principal component index) were obtained in Supplementary Materials S4. Also, we presented a loading plot using the first three principal components (Fig. 6). As seen from the eigenvector coefficients and the loading plot, principal Factor 1, as a dominating extracted component represented the information of $\varphi$Po, $\psi$Eo, $\varphi$Eo, $PI_{ABS}$, $PI_{CS}$ and $PI_{total}$, mainly reflecting the changes in light energy yields, flux ratio and photosynthetic efficiency. Principal Factor 2 comprised Chl *a*, Chl *b* and Caro, explained by photosynthetic pigment factors. Principal Factor 3 mainly included ABS/RC, TRo/RC and ETo/RC, explaining specific energy fluxes per reaction center. Principal component 4 represented plant height, Fo and Fm, depending on the growth, and the initial and maximum fluorescence. Principal component 5 was included by Chl *a*/Chl *b* and Chls/Caro, explained by the composition of chlorophyll and carotenoid.

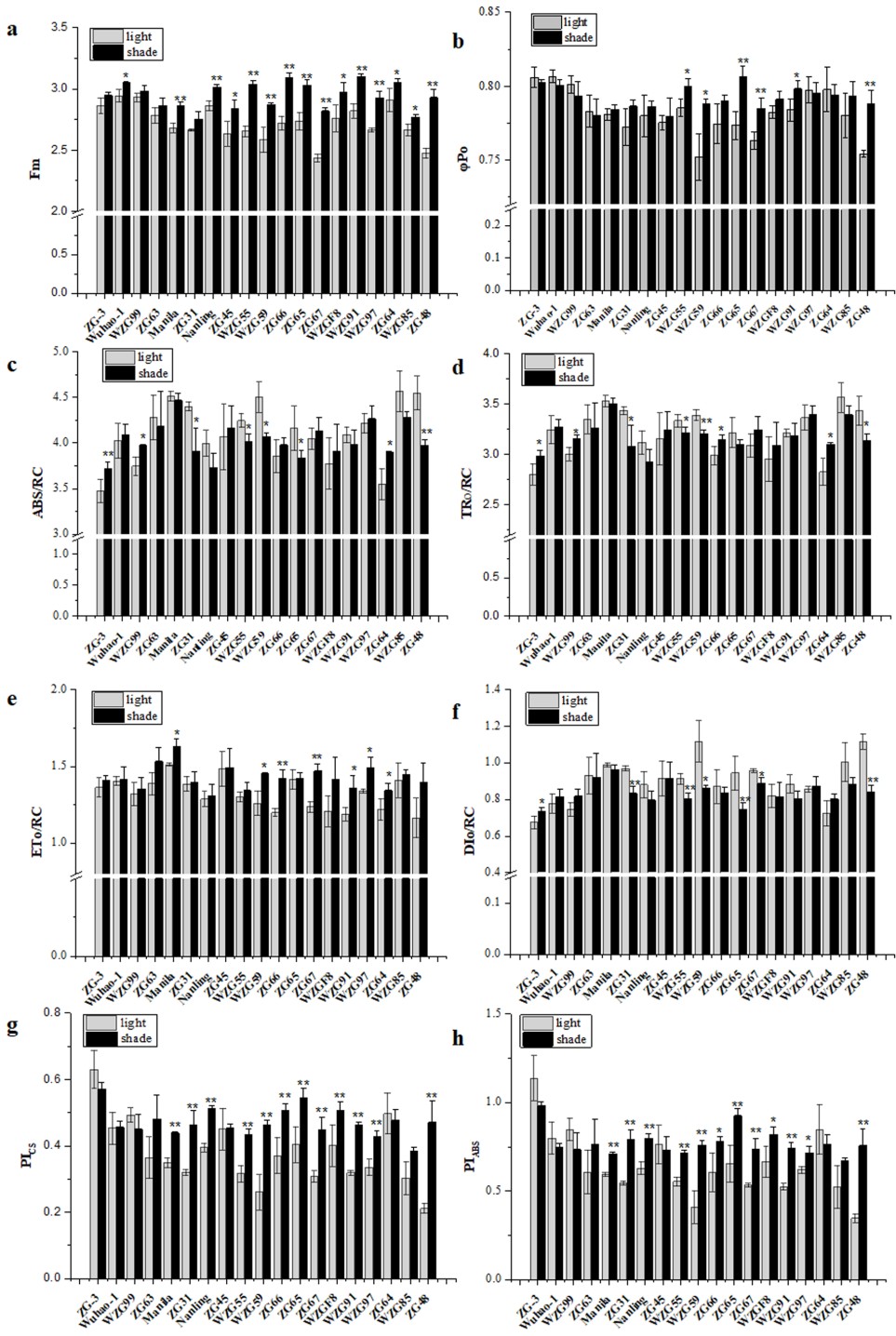

**Figure 5   Changes of prime chlorophyll fluorescence parameters of zoysiagrass after shading, including Fm (A), $\varphi$ Po (B), ABS/RC (C), TRo/RC (D), ETo/RC (E), DIo/RC (F), PI$_{ABS}$ (G), PI$_{CS}$ (H).** Asterisks (* and **) indicate statistically significant differences at $P < 0.05$ and $P < 0.01$ respectively, when indicators in shade were compared with those in light, by SPSS version 19.0 software at the independent samples $T$-test.

**Membership function and comprehensive analysis of zoysiagrass**

The membership function values of each test zoysiagrass, U1 ∼U5, were evaluated according to the formula (Table 3). Based on the E values of the five principal factors (8.08, 3.00, 2.41, 1.42 and 1.14), the index weights (W) were calculated to be 0.50, 0.19, 0.15, 0.09, and 0.07, respectively. The comprehensive evaluation values, D, indicated the relative level of the shade response and were computed and ranked. Among them, 'ZG48' had the largest D value of 0.74, which was the variety with the strongest shade avoidance response. 'ZG-3' having the lowest D value of 0.19, was the weakest variety in the shade avoidance response. That is, 'ZG-3' mainly adopted the strategy of shade tolerance reaction to the shading surroundings. Meanwhile, the shading responses of these zoysiagrass accessions were ranked as follows: 'ZG48' > 'WZG59' > 'ZG97' > 'ZG67' > 'ZG66' > 'WZG91' > 'WZGF8' > 'WZG85' > 'WZG55' >'ZG31' > 'Nanling' > 'ZG65' > 'Manila' > 'WZG99' > 'Wuhao-1' > 'ZG45' > 'ZG63' > 'ZG64' > 'ZG-3'.

Using a statistical index of squared Euclidean distance, nineteen shade trial zoysiagrass were clustered into four categories by D values (Fig. 7). Relatively varying values of 'WZG59' and 'ZG48' were greater, belonging to the type of strong shade avoidance reaction, and judged to be category I. 'WZGF8', 'WZG91', 'WZG55', 'WZG85', 'Nanling', 'ZG65', 'ZG31', 'ZG66', 'ZG67', and 'WZG97' belonged to category II, whose shade avoidance responses were above to medium. The relative values of 'ZG63', 'ZG45', 'Wuhao-1', 'WZG99', and 'Manila' were lower to mid degree, classifying to category III. 'ZG-3' and 'ZG64' possessed the smallest avoidance responses, categorized as category IV.

## DISCUSSION

Currently, urban landscaping planning mainly involves trees, shrubs and grass combinations of the three organisms but often brings the shade onto lawns. However, crowded high-rise clusters are another source of shade stress. Turfgrasses live at the bottom of landscape ecosystems suffering from shade, whose survival mechanisms and strategies have been shaped by evolution. In situations of light shortage, different types of turf grasses have evolved to either tolerate or avoid shading caused by nearby competitors. Previous studies have investigated that although *Cynodon dactylon* and *Zoysia* spp. are both widely used warm-season grasses; between them, *Zoysia* spp. had better shade tolerance (*Harivandi et al., 1984*; *Malik et al., 2014*; *Jespersen & Xiao, 2021*). Therefore, zoysiagrass accessions were chosen in this project to systematically probe their underlying photosynthetic characteristic responses under shaded environments.

There are two strategies for plants to deal with shade: shade avoidance and shade tolerance (*Gommers et al., 2013*; *Wen, Liu & Yang, 2019*). When shaded, most species exhibit shade avoidance syndrome, a common strategy to avoid shade, including elongation of stems and petioles, upward movement of the leaves, and reduced leaf angles (*Franklin, 2008*; *Roig-Villanova & Martinez-Garcia, 2016*). In contrast, some species, subsisting in the understory, which have adapted their morphogenesis and inner adjustment to cope permanently with shaded environments, are inclined to a shade-tolerant response (*Valladares & Niinemets, 2008*). For lawn grasses, which cannot outgrow the heights of

Xu et al. (2022), *PeerJ*, DOI 10.7717/peerj.14274

Peerj

**Table 1 Shade response coefficients correlation matrix of each zoysiagrass photosynthetic index.**

| Individual index | Plant height | Chl$a$ | Chl$b$ | Caro | Chl$a$/Chl$b$ | Chls/Caro | Fo | Fm | $\varphi$Po | $\psi$Eo | $\varphi$Eo | ABS/RC | TRo/RC | ETo/RC | DIo/RC | PI$_{ABS}$ | PI$_{CS}$ | PI$_{total}$ |
|---|---|---|---|---|---|---|---|---|---|---|---|---|---|---|---|---|---|---|
| Plant height | 1 | | | | | | | | | | | | | | | | | |
| Chl$a$ | 0.612[**] | 1 | | | | | | | | | | | | | | | | |
| Chl$b$ | 0.346[*] | 0.826[**] | 1 | | | | | | | | | | | | | | | |
| Caro | 0.587[**] | 0.988[**] | 0.849[**] | 1 | | | | | | | | | | | | | | |
| Chl$a$/Chl$b$ | 0.473[**] | 0.210 | −0.307[*] | 0.158 | 1 | | | | | | | | | | | | | |
| Chls/Caro | 0.384[**] | 0.607[**] | 0.624[**] | 0.530[**] | −0.004 | 1 | | | | | | | | | | | | |
| Fo | −0.008 | −0.064 | −0.144 | −0.053 | 0.003 | −0.244 | 1 | | | | | | | | | | | |
| Fm | 0.268[*] | 0.182 | −0.026 | 0.152 | 0.273[*] | 0.128 | 0.332[*] | 1 | | | | | | | | | | |
| $\varphi$Po | 0.233 | 0.302[*] | 0.155 | 0.267[*] | 0.337[*] | 0.359[**] | −0.462[**] | 0.502[**] | 1 | | | | | | | | | |
| $\psi$Eo | 0.261[*] | 0.302[*] | 0.115 | 0.257 | 0.275[*] | 0.366[**] | −0.136 | 0.441[**] | 0.527[**] | 1 | | | | | | | | |
| $\varphi$Eo | 0.275[*] | 0.330[*] | 0.131 | 0.283[*] | 0.310[*] | 0.399[**] | −0.195 | 0.505[**] | 0.665[**] | 0.980[**] | 1 | | | | | | | |
| ABS/RC | −0.148 | −0.150 | −0.164 | −0.132 | −0.052 | −0.363[**] | 0.315[*] | −0.316[*] | −0.607[**] | −0.461[**] | −0.518[**] | 1 | | | | | | |
| TRo/RC | −0.125 | −0.102 | −0.158 | −0.090 | 0.019 | −0.323[*] | 0.244 | −0.225 | −0.451[**] | −0.384[**] | −0.417[**] | 0.976[**] | 1 | | | | | |
| ETo/RC | 0.211 | 0.208 | −0.043 | 0.171 | 0.357[**] | 0.080 | 0.039 | 0.266[*] | 0.196 | 0.595[**] | 0.559[**] | 0.334[*] | 0.435[**] | 1 | | | | |
| DIo/RC | −0.194 | −0.241 | −0.171 | −0.212 | −0.204 | −0.388[**] | 0.416[**] | −0.463[**] | −0.878[**] | −0.536[**] | −0.645[**] | 0.906[**] | 0.810[**] | 0.117 | 1 | | | |
| PI$_{ABS}$ | 0.233 | 0.307[*] | 0.187 | 0.265[*] | 0.224 | 0.450[**] | −0.312[*] | 0.543[**] | 0.828[**] | 0.830[**] | 0.899[**] | −0.785[**] | −0.683[**] | 0.227 | −0.896[**] | 1 | | |
| PI$_{CS}$ | 0.272[*] | 0.357[**] | 0.175 | 0.315[*] | 0.281[*] | 0.424[**] | −0.193 | 0.578[**] | 0.719[**] | 0.917[**] | 0.956[**] | −0.681[**] | −0.587[**] | 0.367[*] | −0.776[**] | 0.959[**] | 1 | |
| PI$_{total}$ | 0.148 | 0.179 | 0.233 | 0.173 | 0.080 | 0.293[*] | −0.239 | 0.356[**] | 0.640[**] | 0.607[**] | 0.648[**] | −0.775[**] | −0.716[**] | 0.015 | −0.785[**] | 0.785[**] | 0.715[**] | 1 |

**Notes.**

The asterisks, * ($P < 0.05$) and ** ($P < 0.01$), indicate significant differences obtained through SPSS version 19.0 software by Spearman correlation.

**Table 2  The principal component matrix of each individual index.**

| Index | Principal component | | | | |
|---|---|---|---|---|---|
| | 1 | 2 | 3 | 4 | 5 |
| Plant height | 0.41 | 0.49 | 0.20 | 0.40 | −0.32 |
| Chl $a$ | 0.47 | 0.86 | −0.04 | 0.10 | −0.04 |
| Chl $b$ | 0.34 | 0.80 | −0.37 | −0.02 | 0.29 |
| Caro | 0.39 | 0.88 | −0.05 | 0.15 | −0.08 |
| Chl $a$/Chl $b$ | 0.27 | 0.08 | 0.62 | 0.26 | −0.60 |
| Chls/Caro | 0.59 | 0.37 | −0.20 | −0.17 | 0.37 |
| Fo | −0.21 | −0.17 | 0.28 | 0.75 | 0.44 |
| Fm | 0.51 | −0.26 | 0.28 | 0.54 | 0.35 |
| $\varphi$Po | 0.82 | −0.10 | 0.02 | −0.08 | −0.13 |
| $\psi$Eo | 0.85 | −0.11 | 0.37 | −0.19 | 0.12 |
| $\varphi$Eo | 0.91 | −0.11 | 0.32 | −0.18 | 0.08 |
| ABS/RC | −0.80 | 0.28 | 0.48 | −0.15 | 0.12 |
| TRo/RC | −0.69 | 0.30 | 0.56 | −0.19 | 0.10 |
| ETo/RC | 0.27 | 0.17 | 0.86 | −0.33 | 0.21 |
| DIo/RC | −0.88 | 0.22 | 0.28 | −0.09 | 0.15 |
| PI$_{ABS}$ | 0.96 | −0.19 | 0.08 | −0.12 | −0.02 |
| PI$_{CS}$ | 0.95 | −0.18 | 0.17 | −0.07 | 0.03 |
| PI$_{total}$ | 0.84 | −0.18 | −0.18 | −0.03 | 0.12 |
| Eigenvalues (E) | 8.08 | 3.00 | 2.41 | 1.42 | 1.14 |
| Variance contribution rate (%) | 44.88 | 16.66 | 13.40 | 7.86 | 6.35 |
| Cumulative contribution rate (%) | 44.88 | 61.54 | 74.94 | 82.80 | 89.16 |

surrounding trees or buildings, shade-tolerant responses are more practical for low-maintenance conditions where mowing is minimized. In this study, the morphological changes of different zoysia grasses to shade were discrepant. The untrimmed heights of 'ZG-3', 'Wuhao-1', 'WZG99', 'ZG63', 'Manila', 'ZG31', 'Nanling', 'ZG45', 'WZG55', 'WZG59', 'ZG66', and 'ZG67' produced no significant increases under interception of approximately 85% sunlight, revealing more pragmatic lawn growth characteristics. 'ZG48', 'WZG85', 'WZG97', 'WZG91', 'WZGF8', and 'ZG65' exhibited quite strong shade avoidance responses, such as increases in plant height, and leaf and internode elongation, which imposed burdens on daily management.

Photosynthetic pigments are vital components of the plant photosynthetic apparatus and play an important role in photosynthesis to adapt to flexible environments. Shading reduces the total amount of radiation reaching the plants' surface. Pigments absorb more light energy, and their abundances depend on shade intensity, duration and species (*Huylenbroeck & Bockstaele, 2001*). Under low light intensities, most turf grasses capture more light energy by increasing the accumulation of photosynthetic pigments (*Zhou et al., 2003*; *Wherley, Gardner & Metzger, 2005*), but if pigment synthesis pathways are damaged or degraded, the contents eventually decrease (*Bell & Danneberger, 1999*; *Zhou et al., 2010*). Carotenoids are beneficial to the absorption of blue violet light and are crucial antioxidants during photosynthesis (*Simkin et al., 2022*). In this study, chlorophyll and
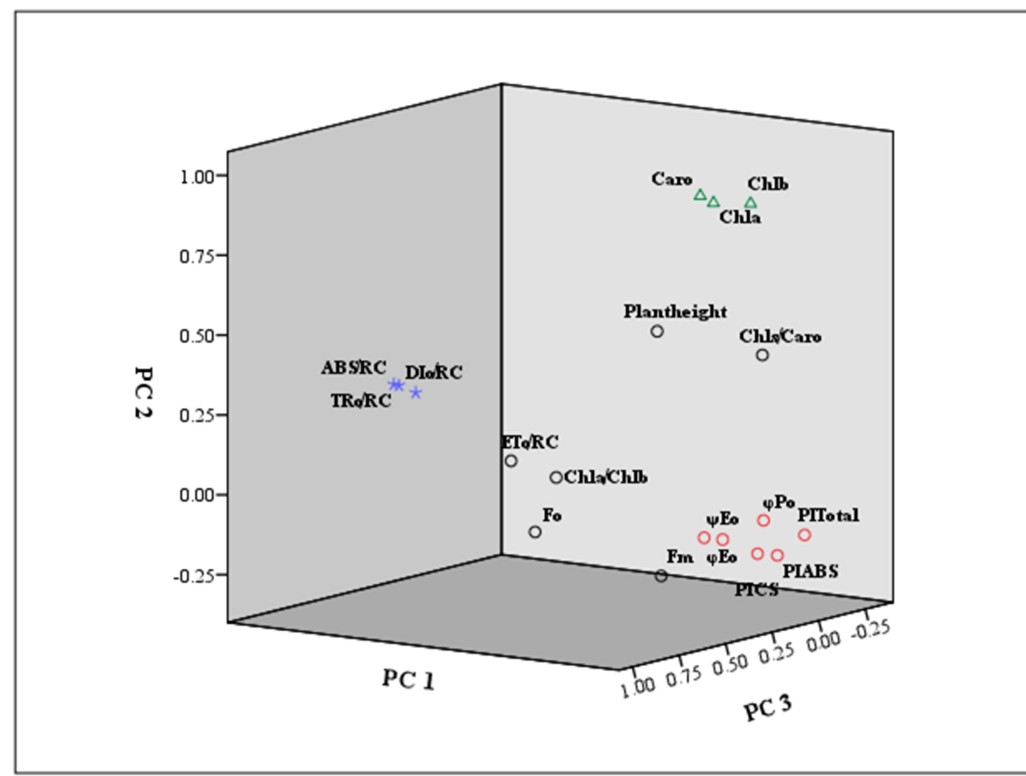

**Figure 6  Loading plot using the first three principal components.** The indicators marked in red square were mainly classified as principal component 1 (PC 1) . The principal component 2 (PC 2) was marked by green triangle. The principal component 3 (PC 3) was signed by purple star.

carotenoid concentrations increased significantly in almost all germplasms, with some increasing highly after 85% shade treatment for 50 d. 'ZG-3' and 'ZG64' maintained strong patience because their chlorophyll and carotenoid contents were relatively high. 'ZG63' shows decreases in chlorophyll and carotenoid concentrations. From the ratios of Chls/Caro, it can be seen that additions of chlorophylls are significantly higher than those of carotenoids in the majority of measurements, apart from 'ZG-3', 'WZG99', and 'ZG63'. In addition, shading alters the qualities of light, and various pigments have peak spectral absorption selectively, resulting in differences in pigment compositions. Some scholars considered that a higher content of Chl *b* may enhance the capture of blue violet wave light, and the Chl *a*/Chl *b* value was used to reflect the shade tolerance of turfgrass, but there was incoherence reported (*Johnson et al., 1993*; *Lichtenthaler et al., 2007*; *Xie et al., 2020*). Among nineteen shade-treated zoysiagrasses we observed, the Chl *a*/Chl *b* values of 'ZG66', 'ZG65', 'WZG97', 'ZG48', and 'WZG85' increased remarkably, 'Wuhao-1' and 'Nanling' decreased observably, and twelve others did not change significantly. These results may infer that changes in pigment concentrations and ratios of Chl *a*/Chl *b* are specific acclimations to different shading conditions of discriminative species, which should not be regarded as a shade-tolerant evaluation index separately.

**Table 3  Scores of principal component factors, the membership function values, index weights and comprehensive valuations of zoysiagrass accessions.**

| Accessions | Y1 | Y2 | Y3 | Y4 | Y5 | U1 | U2 | U3 | U4 | U5 | D value | Comprehensive valuation |
|---|---|---|---|---|---|---|---|---|---|---|---|---|
| ZG-3 | 1.90 | 1.96 | 2.44 | 0.51 | 1.20 | 0.05 | 0.25 | 0.35 | 0.30 | 0.51 | 0.19 | 19 |
| Wuhao-1 | 2.21 | 2.31 | 2.26 | 0.59 | 1.32 | 0.21 | 0.44 | 0.03 | 0.43 | 0.81 | 0.29 | 15 |
| WZG99 | 2.13 | 2.71 | 2.33 | 0.72 | 1.19 | 0.17 | 0.66 | 0.15 | 0.66 | 0.49 | 0.32 | 14 |
| ZG63 | 2.32 | 1.51 | 2.57 | 0.35 | 1.21 | 0.27 | 0.00 | 0.58 | 0.00 | 0.53 | 0.26 | 17 |
| Manila | 2.38 | 1.96 | 2.54 | 0.55 | 1.13 | 0.30 | 0.25 | 0.52 | 0.36 | 0.36 | 0.33 | 13 |
| ZG31 | 2.95 | 1.98 | 2.29 | 0.41 | 1.26 | 0.61 | 0.26 | 0.08 | 0.11 | 0.65 | 0.43 | 10 |
| Nanling | 2.66 | 2.35 | 2.24 | 0.55 | 1.29 | 0.45 | 0.46 | 0.00 | 0.37 | 0.75 | 0.40 | 11 |
| ZG45 | 2.14 | 2.09 | 2.40 | 0.62 | 1.15 | 0.18 | 0.32 | 0.28 | 0.50 | 0.40 | 0.26 | 16 |
| WZG55 | 2.71 | 2.45 | 2.36 | 0.69 | 1.24 | 0.48 | 0.52 | 0.20 | 0.61 | 0.61 | 0.47 | 9 |
| WZG59 | 3.56 | 2.95 | 2.30 | 0.49 | 1.40 | 0.94 | 0.79 | 0.10 | 0.25 | 1.00 | 0.73 | 2 |
| ZG66 | 2.68 | 2.45 | 2.74 | 0.68 | 1.03 | 0.47 | 0.52 | 0.86 | 0.60 | 0.10 | 0.52 | 5 |
| ZG65 | 2.70 | 2.10 | 2.43 | 0.61 | 1.00 | 0.47 | 0.32 | 0.33 | 0.46 | 0.03 | 0.39 | 12 |
| ZG67 | 2.85 | 2.71 | 2.51 | 0.59 | 1.33 | 0.55 | 0.66 | 0.47 | 0.44 | 0.83 | 0.57 | 4 |
| WZGF8 | 2.60 | 2.51 | 2.59 | 0.66 | 1.13 | 0.42 | 0.55 | 0.60 | 0.55 | 0.36 | 0.48 | 7 |
| WZG91 | 2.78 | 2.17 | 2.58 | 0.57 | 1.14 | 0.52 | 0.36 | 0.58 | 0.40 | 0.39 | 0.48 | 6 |
| WZG97 | 2.79 | 3.33 | 2.47 | 0.90 | 1.12 | 0.52 | 1.00 | 0.40 | 1.00 | 0.34 | 0.62 | 3 |
| ZG64 | 1.81 | 1.83 | 2.61 | 0.45 | 1.29 | 0.00 | 0.17 | 0.64 | 0.18 | 0.74 | 0.20 | 18 |
| WZG85 | 2.74 | 2.64 | 2.40 | 0.74 | 0.98 | 0.50 | 0.62 | 0.27 | 0.71 | 0.00 | 0.47 | 8 |
| ZG48 | 3.68 | 1.92 | 2.82 | 0.55 | 1.05 | 1.00 | 0.22 | 1.00 | 0.36 | 0.16 | 0.74 | 1 |
| Index weight (W) | | | | | | 0.50 | 0.19 | 0.15 | 0.09 | 0.07 | | |

In addition, plants might optimize the light capture efficiencies and reactions of the photosynthetic apparatus to accommodate the adjustment and decrease in the quality and quantity under low light solar radiation conditions. It is very important to perform photosynthesis measurements to inspect their photosynthetic physiological adaptation. Even though it does not represent the entire photosynthetic chain, chlorophyll fluorescence could be used to estimate the efficiency of the preliminary steps of photosynthesis, including light energy harvesting, absorption, transmission, and dissipation associated with PSII (*Baker, 2008*). Thus, it accurately reflects the response of plants to changes in their habitat conditions, including the shade environment, which has been widely applied (*D̨abrowski et al., 2015*; *Guidi, Lo Piccolo & Landi, 2019*; *Jespersen & Xiao, 2021*). In our study, the chlorophyll fluorescence kinetic curves and fluorescence parameters of most accessions were significantly increased in response to shading. We observed that among the nineteen zoysia accessions, the indices $\varphi Po$, $\psi Eo$, $\varphi Eo$, ETo/RC, $PI_{ABS}$, $PI_{CS}$, and $PI_{total}$ of some accessions, such as 'ZG-3' 'Wuhao-1' and 'WZG99' were maintained at high levels under light but did not decrease significantly after shading. The light energy absorbed by the PS II unit reaction center (ABS/RC) increased markedly, the majority of the captured light energy used for reduction $QA^−$ (TRo/RC) added significantly, and the energy used for electron transport (ETo/RC) increased slightly, but a small amount of energy was dissipated (DIo/RC), suggesting that they were very patient in shading surroundings. Some ones, for

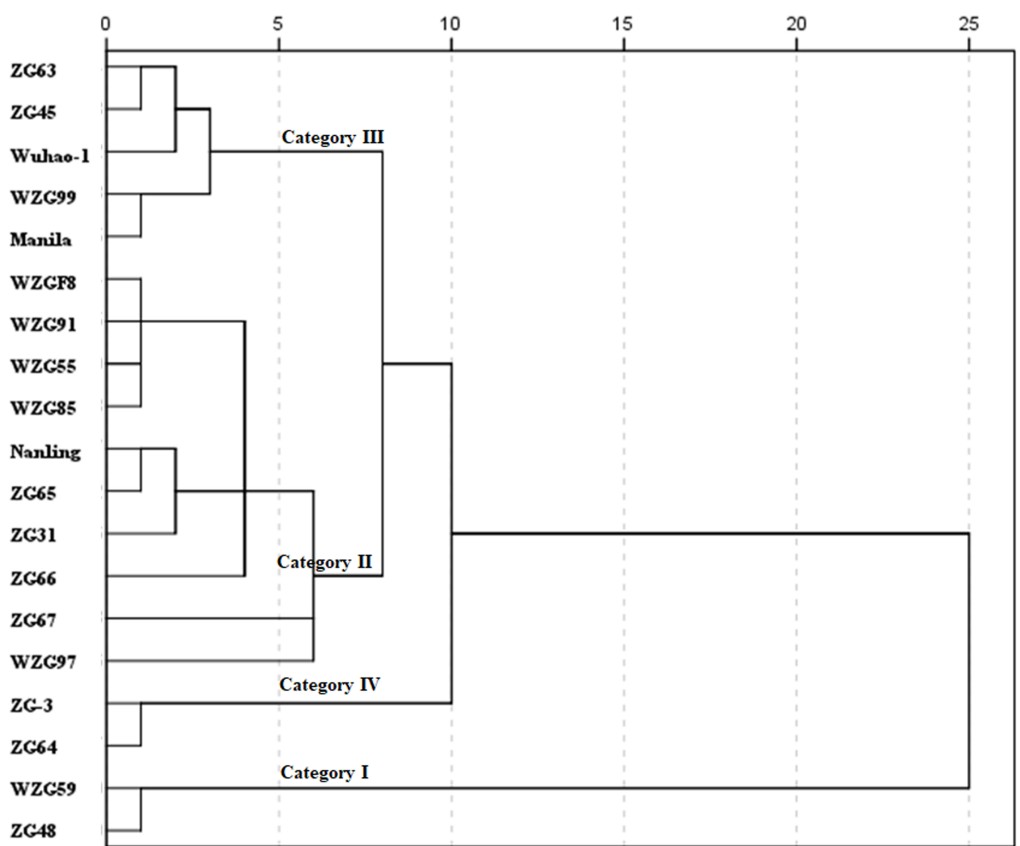

**Figure 7** **Cluster diagram of comprehensive shade response of nineteen zoysiagrass.** The distance between D values was measured by nearest neighbor element analysis model for IBM SPSS Statistics, according to the squared Euclidean distance.

example, 'ZG48', 'WZG59', and 'WZG97' had relatively low indices under light, which increased significantly after shading, promoting their photosynthetic maximum quantum yields, efficiencies and capacities, but most of them still did not reach the level mentioned above. The light energy absorbed by the PSII unit reaction center decreased significantly, the energy used for QA⁻ reduction was observably reduced, the electron transfer energy increased dramatically, and the energy dissipation lessened significantly.

In brief, the photosynthetic system parameters of zoysia were affected by shade treatment to varying degrees, and the role of each index in the shade response might be inconsistent. It is one-sided and confusing to judge shade avoidance or shade tolerance resources by multiple indicators accurately. Therefore, more statistical methods were applied to integrate the index information to comprehensively evaluate the shading response of each accession. PCA has the advantages of eliminating the correlation among indicators, avoiding information duplication and simplifying data analysis (*Sulistyowati et al., 2016*; *Gratani, Vasheka & Puglielli, 2019*). The membership function method is based on the relative shade response values that can eliminate the inherent differences among different germplasms (*Wan et al., 2020*). PCA combined with the membership function method has been used

to evaluate the shade tolerance of plants soybean (*Chunhong et al., 2014*), potato (*Liu et al., 2019a*; *Liu et al., 2019b*), and peanut (*Shijie et al., 2021*). In our research, multinomial photosynthetic indices related to shade were converted into five principal factors by PCA. On this basis, the weight of each new factor was determined, and comprehensive evaluation values (D values) were calculated by the membership function analysis method.

## CONCLUSIONS

In this study, nineteen zoysia grasses were treated with 85% artificial shading, and a large number of growth and photosynthetic characteristic indices under shaded and full sun environments were obtained. Then, they were divided into four categories based on comprehensive evaluations by PCA, membership function analysis and D-value cluster analysis. Category I, 'WZG59', and 'ZG48' adopted a strong shade-avoidance strategy by significantly increasing plant height, photosynthetic pigment contents, fluorescence kinetic curves and photochemical parameter responses to shading surroundings. In category II, 'WZGF8', 'WZG91', 'WZG55', 'WZG85', 'Nanling', 'ZG65', 'ZG31', 'ZG66', 'ZG67', and 'WZG97' the moderate shade avoidance strategy was taken, and the shade avoidance indices were medium. Category III, 'ZG63', 'ZG45', 'Wuhao-1', 'WZG99', and 'Manila', tended to choose a degree of shade-tolerant strategy. The index values of category IV, 'ZG-3' and 'ZG64' were at very high levels and did not change basically after shading, with very strong photosynthetic physiological endurance to be employed in shade tolerance. According to these methods, we deeply understood the shade avoidance and shade tolerance response of zoysiagrass. Additionally, the evaluation system can be used to rapidly evaluate and predict the shade response of grass.

### Abbreviations

| | |
|---|---|
| **ABS/RC** | Absorption flux per unit reaction center |
| **Caro** | Carotenoid |
| **Chl** | Chlorophyll |
| **DIo/RC** | Dissipated excitation energy flux per unit reaction center |
| **ETo/RC** | Electron transport flux per unit reaction center |
| **Fo** | Minimum PSII fluorescence yield at open centers |
| **Fm** | Maximum PSII fluorescence yield at closed centers |
| **KMO** | Kaiser-Meyer-Olkin |
| **PCA** | Principal Component Analysis |
| **PSII** | Photosystems II |
| **PI$_{ABS}$** | Performance index based on absorbed light energy |
| **PI$_{CS}$** | Performance index based on unit cross-sectional area |
| **PI$_{total}$** | Measuring the performance up to the PSI end electron acceptors |
| **TRo/RC** | Maximal trapping flux per unit reaction center |
| **$\Phi$Po** | Maximum quantum efficiency of PSII |
| **$\Psi$Eo** | Efficiency that an electron moves further than QA$^-$ |
| **$\Phi$Eo** | Quantum yield for electron transport |

### Funding

This work was supported by National Key R&D Program of China (2019YFD0900702) and Agricultural Variety Improvement Project of Shandong Province (No. 2019LZGC010). The funders had no role in study design, data collection and analysis, decision to publish, or preparation of the manuscript.

### Grant Disclosures

The following grant information was disclosed by the authors:
National Key R&D Program of China: 2019YFD0900702.
Agricultural Variety Improvement Project of Shandong Province: 2019LZGC010.

### Competing Interests

The authors declare there are no competing interests.

### Author Contributions

- Xiao Xu performed the experiments, analyzed the data, prepared figures and/or tables, authored or reviewed drafts of the article, and approved the final draft.
- Hongli Wang performed the experiments, prepared figures and/or tables, and approved the final draft.
- Guangyang Wang analyzed the data, prepared figures and/or tables, and approved the final draft.
- Xiaoning Li analyzed the data, prepared figures and/or tables, and approved the final draft.
- Xiaoyan Liu analyzed the data, prepared figures and/or tables, and approved the final draft.
- Jinmin Fu conceived and designed the experiments, authored or reviewed drafts of the article, and approved the final draft.

### Data Availability

The raw measurements are available in the Supplementary Files.

### Supplemental Information

Supplemental information for this article can be found online at http://dx.doi.org/10.7717/peerj.14274#supplemental-information.

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
