# Peer review of "Different photosynthetic adaptation of Zoysia spp. under shading: shade avoidance and shade tolerance response"

_PeerJ, doi:10.7717/peerj.14274_

## Round 0.1 · original submission · Major Revisions

Noble Authors,

Three independent experts gave their opinion on your work. Everyone agreed that the work could be published in PeerJ, but had to be significantly corrected beforehand. Please read the opinions of the reviewers and respond to all comments.

With regards,

Reviewer 1 ·

Basic reporting

Use it by author langauge is corect and confirm profesional writing standarts. Used reference proprly describe backgroud of the problem presented in the work

Experimental design

Experiment design is intereresting but must supplemented by adding some iformation listed in detailed review

Validity of the findings

There is lack of information in how many replications the measurements and obervations were made by authors but the final conclusions they made is interesting

Additional comments

Interestin work but for finalizing publish process some corrections are necessary

Annotated reviews are not available for download in order to protect the identity of reviewers who chose to remain anonymous.

Reviewer 2 ·

Basic reporting

English language in this manuscript needs to be extensively improved. Authors may find colleagues who are proficient in English review your text or search for a professional editing service.
Sufficient background information was provided.
This manuscript has professional article structure, figures, and tables. Thanks for sharing your raw data.
There was no reference for the definition of D value. Did authors create the evaluation system? If yes, how did you evluate if it works?

Experimental design

Line 27-30: In this study, 85% shading treatment was applied to nineteen zoysiagrass genotypes, morphological observations and extensive determinations on plant heights, photosynthetic pigments, fluorescence dynamic curves and parameters were made.
Line 44-51: Abbreviations can be listed in a table to make them be easy to search.
Line 63: turfgrasses
Line 68: There should not be a spacing between number and percent sign. Please also check in other lines.
Line 68-69: It is estimated that 20% to 25% of grass in the U.S. and 50% of turf in China are shaded growing in shaded areas.
Line 70, 326: You may use ‘solar radiation’ instead of ‘light quantum’. Please also check in other lines.
Line 71-73: English need to be improved here.
Table S1: Origin of the 19 accessions was unclear. In line 121, you mentioned ‘origin regions’, but in the table, some of the origins were not regions, if the origin is another accession, you may include the origin regions of that accession. For regions in China, you may include country name after the province name.
Line 127: Shading treatments were applied in the greenhouse or outside?
‘Shad treatment’ or ‘shade treatment’? No matter which one you choose, please be unified in the whole manuscript.
Line 130: Shade cloth was removed until…
Line 131: How long was the shading treatment applied? I found the answer in line 184, but you should also mention it here.
Line 135: How was the leaf length measured? How many leaves were involved, what kind of leaf is involved? I have similar questions on other parameters.
Line 146-150: You may explain Chl a, Chl b, Chls, and Caro.
Line 174: For the definition of D, please provide reference.
Line 176: What is the meaning of ‘lines’ here? I found similar term in Table S1. Do you mean genotype, accession, or variety? Please be unified in the whole manuscript.

Validity of the findings

Line 185: show instead of occur
Line 206-208: What parameters were these genotypes similar in?
Line 220-223: Did you do any statistical separation to found ‘ZG48’ and ‘ZG-3’ as the most and least changes?
Table 1: You need to explain the meaning of asterisk in the table.
Line 264: What is KMO value?
Line 271-299: How Y1-Y5 were calculated is not necessary in this manuscript, you may include them in the supplementary materials. Since you have done principal component analysis (PCA), why not present a PCA plot using first 2 or 3 principal components?
Line 324: Turfgrasses live at the bottom of landscape ecosystem suffering from
Line 329: delete reversely
Line 330: chosen
Line 407-408: decapitalize Soybean, potato, and peanut
Line 269, 409: principal component instead of comprehensive factor
Line 410: Pay attention to the definition of D value.
Line 410-416: This part looks more like results.
Line 420: full sun instead of light

Reviewer 3 ·

Basic reporting

Manuscript title: Different photosynthetic physiological adaptation of Zoysia spp. under shading: shade avoidance and shade tolerance response

1. General comments
Shade tolerance and avoidance by plant organisms has been an interesting scientific area still in evolution. The illustration is this manuscript that tried to elucidate the response to light limitation of 19 accessions/varieties/species of the genus Zoysia. I thank the authors for this effort. Nevertheless, I noticed as detailed below through my specific comments a number of limitations to this study that deserved to be addressed.

Basically, I had hard time going through the manuscript because of the language. I then insist that the manuscript be deeply edited by an editing house or by a fluent English speaker to raise it to the required standard. For instance:
line 81: Replace “shade-tolerance: by “shade-tolerant” ;
line 138: Do not use plural there, but rather singular: “height” and not “heights”, “thickness” and not “thicknesses”, etc;
line 155: “Comprehensively” should rather read “Comprehensive”;
line 226: “ the increase occurred” and not “ the increase was occurred”;
Line 332: “survivorship/survival” and not “survivability”;
Line 338: “have been chosen” and not “have been chose”;
Line 69: Across the manuscript, make sure the percent symbol is always stuck to the figure. “20%” and not “20 %”. Etc.
I am not sure I could capture any hypothesis from the introduction.

Title
The manuscript title needed to be revised to only contain either photosynthetic or physiological, not both, as the photosynthetic mechanism is fully part of the physiological process of the plant.

Abstract
Line 24-26: Rephrase the sentence to make it clearer
Line 26: Replace “our” by “an” and delete “physiological”
Introduction
Lines 55-58. References are needed to support the statements made.
Line 86: Elaborate in one sentence, what TPI measures

Experimental design

2- Materials and methods

*When and where was the study conducted should be clearly indicated.
*I have a serious concern regarding the genetic nature of the plant material used:
Lines 123-125: How many species are the author dealing with? Are authors dealing with varieties or accessions?
Lines 126-129: Author should clearly indicate and describe the experimental design used.
Line 130: What are these treatments?
Lines 133-134: The shading treatment applied is odd. The shading exposure duration applied was determined by signal of phenotypic differences. By so doing, different shade durations were applied by the author and this could have undoubtedly created a bias in the results, and the conclusions are then questionable.
Line 172: I fear Pearson correlation is not suitable for all pairwise correlation? The only condition where authors can perform a Pearson correlation is that all variables are normally distributed. Authors need to check for data normality first, and then use alternative method (e.g., Spearman correlation) in case either of the variables in play was not normal.

At what stage of the experimet did he authors collect the data?

Validity of the findings

I do not see the necessity to consider up to five components since even the first two components explained more than 50% of the variance. I would advise not to go beyond three components. This is ease the data interpreation.

It will be good to directly indicate on the dendrogram the category out of the four identified) to which each genotype falls in.


As indicated above, I have a reservation regarding the differential shade exposure duration applied as this could have a deep implication on the results and conclusions as well.

---

## Round 0.2 · Minor Revisions

Dear Authors,

One Reviewer has two remaining questions for this version of your work. Please read and respond to these comments.

With best regards,

Reviewer 2 ·

Basic reporting

I accept most of the responses to my comments. Except for the following questions:

Lines 141-145: I didn’t find answers for my question ‘Shading treatments were applied in the greenhouse or outside?’ Besides, how did authors get the average ‘738 μmol m−2 s−1 of photosynthetically active radiation’ (PAR)? How many times were PAR measured, on what time?

Lines 159-163: Again, how many plants were included in the measuring of these data? How many plants were included in measurement within each experimental unit has statistical meanings, it should not be skipped with ‘data were not exhaustively counted’. Besides, I didn’t find results of leaf length, stem thickness, and whether they were easy to be lodged. If these results were not presented in this manuscript, they should be deleted in materials and methods.

Experimental design

no comment

Validity of the findings

no comment

---

## Round 0.3 · accepted · Accept

Dears Authors,

The reviewer had no doubts about your work. Therefore, I have made the decision to publish it in the current version. My congratulations!

With regards,

The Section Editor added some suggested edits:

EDITS LINE NO: / BEFORE / AFTER / [COMMENTS]

LINE 21: / Reducing / Reduction / [.]
LINE 29: / curves and / curves among other / [.]
LINE 61: / building or so, / buildings and such, / [.]
LINE 61: / reduces frequently / is frequently reduced / [.]
LINE 90: / , but some / , but with some / [.]
LINE 95: / that occurred / that germplasm occurred / [.]

Reviewer 2 ·

Basic reporting

I accept the responses to my comments. I would like to recommend this manuscript be accepted for publishing.

Experimental design

no comment

Validity of the findings

no comment